

# Efficacy of propolis extract and eye drop solutions to suppress encystation and excystation of *Acanthamoeba triangularis* WU19001-T4 genotype

Suthinee Sangkanu[1], Abolghasem Siyadatpanah[2], Roghayeh Norouzi[3], Julalak Chuprom[4], Watcharapong Mitsuwan[5], Sirirat Surinkaew[6], Rachasak Boonhok[6], Alok K. Paul[7], Tooba Mahboob[8], Imran Sama-ae[9], Sonia M. R. Oliveira[10,11], Tajudeen O. Jimoh[12,13], Maria de Lourdes Pereira[10,14], Polrat Wilairatana[15], Christophe Wiart[16], Mohammed Rahmatullah[17], Monvaris Sakolnapa[18] and Veeranoot Nissapatorn[1]

[1] School of Allied Health Sciences, Southeast Asia Water Team (SEA Water Team), World Union for Herbal Drug Discovery (WUHeDD), and Research Excellence Center for Innovation and Health Products (RECIHP), Walailak University, Nakhon Si Thammarat, Thailand
[2] Department of Microbiology, Gonabad University of Medical Sciences, Gonabad, Iran
[3] Department of Pathobiology, Faculty of Veterinary Medicine, University of Tabriz, Tabriz, Iran
[4] School of Languages and General Education, Walailak University, Nakhon Si Thammarat, Thailand
[5] Akkhraratchakumari Veterinary College and Research Center of Excellence in Innovation of Essential Oil, Walailak University, Nakhon Si Thammarat, Thailand
[6] Department of Medical Technology, School of Allied Health Sciences, Walailak University, Nakhon Si Thammarat, Thailand
[7] School of Pharmacy and Pharmacology, University of Tasmania, Hobart, Australia
[8] Department of Pharmaceutical Biology, Faculty of Pharmaceutical Sciences, UCSI University, Kuala Lumpur, Malaysia
[9] Department of Medical Technology, School of Allied Health Sciences and Center of Excellence Research for Melioidosis and Microorganisms, Walailak University, Nakhon Si Thammarat, Thailand
[10] CICECO-Aveiro Institute of Materials, University of Aveiro, Aveiro, Portugal
[11] Hunter Medical Institute, Hunter Medical Research Research Institute, New Lambton, New South Whales, Australia
[12] Department of Pharmacognosy and Pharmaceutical Botany, Faculty of Pharmaceutical Sciences, Chulalongkorn University, Bangkok, Thailand
[13] Department of Biochemistry, Faculty of Health Sciences, Islamic University in Uganda, Kampala, Uganda
[14] Department of Medical Sciences, University of Aveiro, Aveiro, Portugal
[15] Department of Clinical Tropical Medicine, Faculty of Tropical Medicine, Mahidol University, Bangkok, Thailand
[16] Institute for Tropical Biology & Conservation, University Malaysia Sabah, Sabah, Malaysia
[17] Department of Biotechnology & Genetic Engineering, University of Development Alternative, Dhaka, Bangladesh
[18] School of Medicine, Walailak University, Nakhon Si Thammarat, Thailand

Corresponding author
Veeranoot Nissapatorn,
nissapat@gmail.com

## ABSTRACT

**Background:** Propolis is a natural resinous product from a variety of plants and combines it with beeswax and salivary enzymes to create bee glue. Its potentials have been employed in the treatment of many diseases and conditions for centuries. This study, therefore, aims to determine the anti-*Acanthamoeba* properties of the propolis

extract, eye drops coupled to some mechanisms such as inhibition of encystation and excystation.

**Methods:** The minimum inhibitory concentrations (MICs) of the most active propolis extract were assessed in trophozoites and cysts of *Acanthamoeba triangularis* (*A. triangularis*) at 0.256 and 1 mg/mL, respectively.

**Results:** Active eye drops inhibited trophozoites and cysts with a MIC value of 100%. At concentrations lower than their MICs values (1/2–1/16 MIC), propolis and eye drops revealed inhibition of encystation. In addition, at 1/2 MIC, both the propolis extract and eye drops showed potential inhibition of excystation. In combined sets of propolis extract and eye drops, they showed slightly increasing inhibition of encystation and excystation. Surprisingly, the MIC of chlorhexidine decreased when combined with the propolis and eye drops. SEM and TEM images displayed morphological changes in *A. triangularis* treated with combined sets of propolis extract and eye drops.

**Conclusion:** From this study, a new combined eye drop solution with propolis extract was found to be helpful in reducing encystation and excystation in *A. triangularis*. Therefore, this combination is an alternative for the treatment of eye diseases in early stages along with reducing the entry into the cyst stage of *Acanthamoeba*. The results of this study revealed new anti-*Acanthamoeba* inhibitors with promising combined alternative therapeutic potential for the treatment of *Acanthamoeba* keratitis.

**Subjects** Biochemistry, Microbiology, Drugs and Devices, Infectious Diseases, Ophthalmology
**Keywords** Anti-*Acanthamoeba*, Propolis extract, Encystation, Excystation, Propolis and eye drops

# INTRODUCTION

*Acanthamoeba* spp. are free-living amoebae capable of causing fatal granulomatous amoebic encephalitis (GAE), *Acanthamoeba* keratitis (AK), *Acanthamoeba* pneumonia (AP), cutaneous acanthamoebiasis, and disseminated acanthamoebiasis in humans (*Kot, Łanocha-Arendarczyk & Kosik-Bogacka, 2021*). In healthy individuals with contact lenses, *Acanthamoeba* infection in the eyes has become a great concern in public health worldwide (*Pinto et al., 2021*). According to *Acanthamoeba* life cycle, two parasite stages, trophozoite and cyst, are involved. Trophozoite is a vegetative stage, which survives in neutral pH, adequate food supply, ambient temperature and balanced osmolality, while the cyst can survive in harsh conditions such as lack of food, extreme pH or temperature, and hypo-osmolality. Regarding the *Acanthamoeba* keratitis, the cyst form can be found in an acceptor cornea, and with this form, it becomes a barrier for AK treatment.

The currently recommended treatment strategy is to combine standard anti-*Acanthamoeba* drugs, biguanide and diamidine, for an effective treatment against cysts (*Khan, 2006*). Clinical reasoning processes are difficult to assess AK because the early stage of AK shares other common features with other types of keratitis diseases (viral, bacterial and fungal) such as eye pain and redness (*Turner et al., 2000*). Most patients with keratitis are treated with eye drops. Therefore, if the eye drops do not contain ingredients that can

inhibit or kill *Acanthamoeba* it can cause the spread of infection and takes a long time to treat. As a result, *Acanthamoeba* may become resistant to drugs and return to the cyst stage. Thus, eye drops containing active ingredients are required for early treatment (*Sifaoui et al., 2018*).

Propolis or bee glue is a mixture of honeybees and natural products of different parts of plants (*Pasupuleti et al., 2017*), used for the construction and repairing beehives. Propolis hardens the cell wall of beehives, contributes to an aseptic internal environment (*Shehu et al., 2016*), and acts as a protective barrier against predators. In addition, propolis have several biological activities such as anti-inflammation, anti-proliferation, antioxidant, anti-diabetic activity, and antimicrobial activities (*Przybyłek & Karpiński, 2019*; *Shi et al., 2019*; *Hwang et al., 2020*). Some reports have demonstrated the anti-*Acanthamoeba* activity. In 2007, *Vural et al. (2007)*, reported the efficacy of propolis *in vivo*, found that the keratitis grade on day 10 in the eyes of rats on propolis, chlorhexidine and propolis plus chlorhexidine was significantly lower compared to control eyes ($P < 0.05$). Recently, our study demonstrated the potential ability of the pinocembrin, chrysin, and tectochrysin in propolis extract complex to form hydrogen bonds with the AcSir2 protein which expression is essential for the growth and encystation of *A. castellanii* (*Sama-Ae et al., 2022*). Therefore, this study sought to evaluate an amoebicidal activity and anti-*Acanthamoeba* encystation or excystation by propolis extract and eye drops on *A. triangularis* that could offer an alternative treatment strategy for *Acanthamoeba* infection.

Polyphenols, flavonoids, aromatic and aliphatic acids, lignans, stilbenes, tannins, terpenes, polysaccharides, and aldehydes are abundant in the chemical constitution of propolis extracts (*Aminimoghadamfarouj & Nematollahi, 2017*). Propolis must be treated using suitable solvents to remove the inert material and preserve compounds of interest (*Ahangari, Naseri & Vatandoost, 2018*). Ethanol stands out among other solvents because it produced a propolis extract high in bioactive chemicals with minimum wax content (*Marcucci, 1994*). *Afrouzan et al. (2017)* found that Iranian propolis extract in 70% ethanol was effective against *P. falciparum* 3D7 chloroquine-sensitive strain (CQ) *in vitro* and *P. berghei* (ANKA line) (*in vivo*). After 14 days of infection, the anti-activity of the EtOH 70% fraction led to reductions in parasitemia of 65.1%, 66%, and 71% subsequent treatment with 50–100 and 200 mg/kg of propolis, respectively (*Afrouzan et al., 2017*). Like the methanolic extract of propolis from Saudi Arabia, this one demonstrated *in vivo* antiplasmodial efficacy against *P. chabaudi* and a splenoprotective effect, with parasitemia reductions between 49% and 70%. Treatment with propolis at a dose of 100 mg/kg relieved in the restoration of the spleen gross morphology (*AlGabbani et al., 2017*).

Although several studies with propolis against parasites have been carried out, few have focused on the *Acanthamoeba triangularis*, which causes keratitis. Therefore, we tested the amoebicidal activity and anti-*Acanthamoeba* encystation or excystation of propolis extract and eye drops as this is an important stage in the life cycle of *Acanthamoeba* infection that is involved in the disease. This could offer an alternative treatment strategy for *Acanthamoeba* infection.

## MATERIALS AND METHODS

### Preparation of propolis extracts

Ten samples of propolis were collected from different geographic regions of Iran (Table S1) with the permission and verification of plants from agricultural organizations in respective locations. The ethanolic extract of the plant was prepared by soaking 370 g of dry powder with 1 L of absolute ethanol for 7 days at room temperature. Afterwards, materials were filtered through Whatman filter paper No. 1 (GE Healthcare, Buckinghamshire, UK), the solvent was removed with a rotary evaporator (G3 Heidolph, Burladingen, Germany). Dried extracts were preserved at 4 °C and re-suspended in dimethyl sulfoxide (DMSO) at a 100 mg/mL concentration before use.

### Eye drop solutions

Four patented commercially available eye drops solutions (EDS-1, -2, -3, and -4) for topical use against eye diseases were selected for analysis in this study. Details of the eye drops compositions are shown in Table S2.

### Culture of *A. triangularis* WU19001 belonging to T4 genotype

*A. triangularis* WU19001 belonging to the T4 genotype (MW647650.1) was originally isolated from a recreational reservoir at Walailak University, Thailand. Trophozoites were grown in 75 cm$^2$ tissue culture flasks in PYG medium (proteose peptone 0.75% (w/v), yeast extract 0.75% (w/v) and glucose 1.5% (w/v)) without shaking at 28 °C as described previously (*Mitsuwan et al., 2020*). For cysts, trophozoites were transferred from the PYG medium to the Neff's encystment medium (NEM; 0.1 M KCl, 8 mM MgSO$_4$·7H$_2$O, 0.4 mM CaCl$_2$·2H$_2$O, 1 mM NaHCO$_3$, 20 mM ammediol) and were cultured in this medium for 7 days. To obtain only mature cysts, culture cells were collected and then centrifuged at 4,000 rpm for 5 min. Supernatant was discarded and re-suspended with sodium dodecyl sulfate (SDS) solution at a final concentration of 0.5% (w/v) for 20 min to solubilize trophozoites and immature cysts (*Moon et al., 2014*). Then, mature cysts were harvested and washed twice using phosphate-buffer saline (PBS). All experiments were performed in line with the biosafety guidelines for scientific research at Walailak University, Thailand (Ref. No. WU-IBC-66-020).

### Screening for anti-*Acanthamoeba* activity

All extracts were screened for anti-*Acanthamoeba* activity against *A. triangularis* WU19001 at a final concentration of 1 mg/mL by a broth microdilution method (*Sangkanu et al., 2021*). The stock solution (100 mg/mL) of the extract was prepared in DMSO and diluted to a concentration of 2 mg/mL with PYG medium. Aliquots of 100 μL were transferred to each well in triplicate. To prepare the final 1 mg/mL concentration, 100 μL of trophozoites or cysts ($2 \times 10^5$ cells/mL) were added. Plates were incubated at 28 °C for 24 h.

The percentage of cell viability analysis was the use of 0.2% trypan blue dye exclusion staining and observed under inverted microscopy (Nikon, Tokyo, Japan). The viability of

parasites was calculated as follows: % viability = (mean of the viable parasite/mend of the control) × 100.

## Determination of minimal inhibitory concentration

The minimal inhibitory concentration (MIC) of the active extracts was determined against the *A. triangularis* by broth microdilution methods as described previously (*Mitsuwan et al., 2020*). Briefly, 50 μL of two-fold serial dilutions in PYG medium, of the different extracts were added to each well. After that, 50 μL of *Acanthamoeba* trophozoites and cyst ($2 \times 10^5$ cells/mL) were seeded in the 96-well microtiter plates. The range of tested concentrations was 0.008 to 1 mg/mL. EDS were tested at concentrations of 100% and 50%. Chlorhexidine (Sigma-Aldrich, St. Louis, MA, USA) was tested in range of 0.001–0.064 mg/mL. The lowest concentration of propolis extract or EDS that inhibited 90% of *A. triangularis* growth was recorded as the MIC.

## Inhibition of encystation on *A. triangularis*

Under unfavorable condition, *Acanthamoeba* trophozoites entry to cyst form with metabolically inactive and dormant cysts. Drugs that target *Acanthamoeba* are not effective against these cysts. Therefore, the inhibition of encystation was required. Encystation was performed as previously described (*Dudley Alsam & Khan, 2008*), with modifications. Briefly, propolis extracts or EDS at different concentrations (1/2 MIC, 1/4 MIC, 1/8 MIC, 1/16 MIC) were diluted with Neff's medium in a 96-well plate and then *Acanthamoeba* trophozoites ($5 \times 10^5$ cells/mL) were added. Plates were incubated at 28 °C for 72 h. The total of amoeba number was counted using a haemocytometer (Boeco, Hamburg, Germany), then trophozoites and immature cysts were lysed by adding sodium dodecyl sulfate (SDS; 0.5% final concentration) and incubated for 1 h. After adding SDS, the surviving cysts were counted once again. The percentage of *Acanthamoeba* encystation was calculated as follows: (total number of amoebae post-SDS treatment/total number of amoebae pre-SDS treatment) × 100. *Acanthamoeba* alone served as the negative control, whereas PMSF (10 mM final concentration) was utilized as the positive control.

## Inhibition of excystation on *A. triangularis*

Excystation is the emergence process of trophozoites from cysts resulting in infection recurring. For inhibition of excystation, *Acanthamoeba* cysts ($5 \times 10^5$ cells/mL) were incubated with various concentrations of propolis extracts and EDS (1/2 MIC, 1/4 MIC, 1/8 MIC, 1/16 MIC, 1/32 MIC and 1/64 MIC) in PYG medium in 96-well plate at 28 °C for 72 h. The excystation was determined using a hemocytometer and an inverted microscope. All of amoebae were counted, and then SDS (0.5% final concentration) was added and incubated for 1 h to dissolve the immature cysts and trophozoites. The remaining cysts in the cultures were counted again. The percentage of *Acanthamoeba* excystation was calculated using the following formula: (total number of amoebae pre-treatment with SDS–total number of amoebae post-SDS treatment)/(total number of amoebae pre-SDS treatment) × 100. PMSF (10 mM final concentration) and amoebae alone were used as positive and negative controls, respectively (*Anwar et al., 2019*).

## Effects of combinations on inhibition of encystation and excystation

The propolis extract, alone or in combination with EDS was tested for anti-encystation and excystation. The propolis extract and EDS were diluted with Neff's or PYG medium to obtain four times to their final concentrations of MIC, 1/2 MIC, 1/4 MIC, 1/8 MIC and 1/16 MIC in 96-well polystyrene plates (*Sangkanu et al., 2021*). Then, a total of 100 µL of *Acanthamoeba* trophozoites or cysts ($5 \times 10^5$ cells/mL) was added to each well. Plates were incubated at 28 °C for 72 h. Percentage detections of *Acanthamoeba* encystation and excystation were assessed using previous methods.

## Effects of combination on anti-amoebic activity

The most active combination sets of propolis extract and EDS were integrated with chlorhexidine to determine the anti-amoebic activity. A total of 50 µL chlorhexidine were diluted in 96-well polystyrene plate (four times to their final concentrations of MIC, 1/2 MIC, 1/4 MIC, 1/8 MIC and 1/16 MIC). Then 50 µL of four times to final concentration of combination set was added. A total of 100 µL of trophozoites or cysts ($2 \times 10^5$ cells/mL) was added (*Sangkanu et al., 2021*). The plates were incubated for 24 h at 28 °C without agitation. Cell viability analysis involved dye exclusion staining followed by direct cell counting under the inverted microscopy (Nikon, Tokyo, Japan). Percentage amoebae viability was calculated from (mean of the treated parasite/mean of the control) × 100.

## Calcofluor white staining

The cyst walls of *Acanthamoeba* can be stained blue white in this technique. Then, the results demonstrate that the calcofluor white stain method is a highly reliable technique for identification of *Acanthamoeba* cysts. Then, this technique was used for detection of *Acanthamoeba* cyst after treatment with propolis extract, EDS and combination solution. *Acanthamoeba* cells were incubated under inhibition of encystation condition (propolis extract No. 10, 0.016 mg/mL, and EDS-3, 6.25%) and excystation condition (propolis extract No. 10, 0.512 mg/mL, and EDS-3, 50%) for 72 h. After incubation, *Acanthamoeba* pellets were washed with PBS, re-suspended in a 2.5% calcofluor white staining solution, and incubated for 20 min at room temperature. After washing with PBS, samples were observed under a fluorescence microscope (BX53; Olympus, Tokyo, Japan) (*Moon et al., 2013*).

## Scanning electron microscopy and transmission electron microscopy study

In this study, the morphology and surface of *Acanthamoeba* cells after treatment with combination of propolis extract and EDS was observed by SEM. Combination set of encystation included propolis extract No. 10 (0.016 mg/mL) and EDS-3 (6.25%) and combination set of excystation included propolis extract No. 10 (0.512 mg/mL) and EDS-3 (50%). After incubation for 72 h, parasites were collected by centrifugation at 3,000 rpm for 5 min and re-suspended in PBS. Samples were fixed with 2.5% glutaraldehyde for 2 h and then dehydrated with a series of graded ethanol (20%, 40%, 60%, 80%, 90%, and 100%). Dehydrated samples were mounted on aluminium stubs and allowed to dry using a

critical point dryer EMS/K850 (Quorum, Laughton, UK). Samples were then coated with 99% gold particles with 108 Auto Sputter Coater (Ted Pella Inc., Redding, CA, USA) and the morphology of *A. triangularis* trophozoites and cysts in inhibition of encystation or excystation was subsequently examined under SEM (SEM-Zeiss, Munich, Germany) at the Center for Scientific and Technological Equipment, Walailak University, Nakhon Si Thammarat, Thailand.

Ultrastructure of *Acanthamoeba* after treatment with combined propolis extract and EDS was determined by TEM. Combination set of encystation included propolis extract No. 10 (0.016 mg/mL) and EDS-3 (6.25%) and combination set of excystation included propolis extract No. 10 (0.512 mg/mL) and EDS-3 (50%). Briefly, *Acanthamoeba* cells were harvested by centrifugation at 3,000 rpm for 5 min. The supernatants were discarded and the pellets were washed three times with PBS. The cell pellet was fixed in suspension in 2.5% glutaraldehyde. Post fixed in 1% osmium tetroxide and 2% uranyl acetate, then dehydrated with increasing concentrations of ethanol, and embedded in epoxy resin. After toluidine-blue stained sections, ultra-thin sections of 90 nm thickness were obtained from a diamond knife (Diatome) from the cell blocks, and the sections were stained with a 4% uranyl acetate-lead citrate solution. Cells were examined using the JEM 2010 transmission electron microscope (Jeol, Tokyo, Japan) at an accelerating voltage of 80 kV.

## Cytotoxicity assay

The cytotoxic effects of the most active propolis extract, EDS, and combination sets were evaluated using the Vero cell line (8200F270602; Elabscience, Wuhan, Hubei, China). Vero cells are derived from the kidneys of African green monkeys and are widely used in research because of their adaptability and accessibility (*Jethva & Bhatt Dhara Zaveri, 2016*). The cells were grown in Dulbecco's Modified Eagle's (DMEM) medium (Merck KGaA, Darmstadt, Germany), which was enhanced with 10% FBS and 1% antibiotic supplemented with 100 units/mL of penicillin G and 100 µg/mL of streptomycin. The culture was incubated with 5% $CO_2$ at 37 °C in a humidified environment. At 90% confluence, cells were detached using trypsin and ethylene diamine tetra-acetic acid (EDTA).

A 96-well polystyrene plate was seeded with single cells at a density of $1.5 \times 10^4$ cells/100 µL, and they were left to attach for 24 h. Next, a gentle addition of 100 µL of propolis extract, EDS, and combined set was added. After incubation for 24 h, the cytotoxic effects were determined using the MTT assay (*Wilson et al., 2017*; *Mitsuwan, Wintachai & Voravuthikunchai, 2020*). The absorbance was measured using a microplate reader (Biotek, Cork, Ireland) at 570 nm. The survival percentage was calculated using the following equation:

% survival = (ABt/ABu) × 100.

Cell survival was measured as the percentage absorbance compared with the treated cells (ABt) and non-treated cells (ABu).

## Data analysis

Triplicate of the experiments were carried out. All information was gathered and entered into the statistical package (SPSS Inc., Chicago, IL, USA) software. The data was reported as mean ± SD. To perform statistical analysis, the two-tailed unpaired Student's t-test was employed. In every analysis, a value of $p < 0.05$ was accepted as statistically significant.

# RESULTS

## Anti-*Acanthamoeba* activities

The anti-amoebic activities of propolis extracts on *A. triangularis* trophozoites and cysts were evaluated by calculating the percentage of viability. The activities observed at 24 h were compared to the untreated control, and the standard deviations (SD) are shown in Table S3. The percentage of viability rate of trophozoites in propolis extract from No. 1 (Tabriz city), No. 3 (Pranshahr city), No. 5 (Sarab city) and No. 10 (Kermanshah city) was 0. Furthermore, the cyst viability rate was 5.12 ± 7.94% when treated with 1 mg/mL of propolis extract No. 10. Based on this result, four propolis extracts showed good activity on *A. triangularis* were chosen for MIC determination.

## Determination of the minimum inhibitory concentration

The MIC of active propolis extracts and EDS are summarized in Table S4. The sensitivities of trophozoites were significantly higher than cysts. Propolis extracts No. 1, No. 3, No. 5 and No. 10 exhibited MIC concentration in the trophozoites at 0.128–0.256 mg/mL, whereas propolis extract No. 10 (Kermanshah) showed good anti-*Acanthamoeba* activity in cysts with MIC at 1 mg/mL. Four EDS were also evaluated to determine the MIC concentration against *A. triangularis*. EDS-1, -2 and -3 expressed anti-*Acanthamoeba* activity with MIC at 100%, which inhibited the growth of trophozoites. Moreover, only EDS-3 revealed growth inhibition effects against trophozoites and cysts with MIC at 100%. EDS-4 did not show anti-*Acanthamoeba* activity on trophozoites and cysts. In the positive control, chlorhexidine exhibited MIC values of 0.008 and 0.032 mg/mL for trophozoites and cysts, respectively. Therefore, propolis extract No. 10 and EDS-3 were chosen for further study.

## Inhibition of encystation in *A. triangularis*

To assess the effect of propolis extract No. 10 and EDS3 on *A. triangularis* encystation, PMSF was used as a positive control in Neff's medium. According to the data presented in Fig. 1A, the results revealed that the propolis extract No. 10-exhibited inhibition of *A. triangularis* encystation at all concentrations. At 0.016 mg/mL (1/16 MIC) of propolis extract decreased encystation to 13.58 ± 2.2%. Inhibition of encystation in *A. triangularis* seems to be sensitive to EDS-3 at all concentrations (Fig. 1B). Seventy-two hours after the induction of encystation, the population of mature cysts was mainly present in the untreated control. In the PMSF, propolis extract, and EDS-3 treatments, the trophozoites were detached from the well and started to round and some cells were inverted into mature cysts (Fig. S1).

 

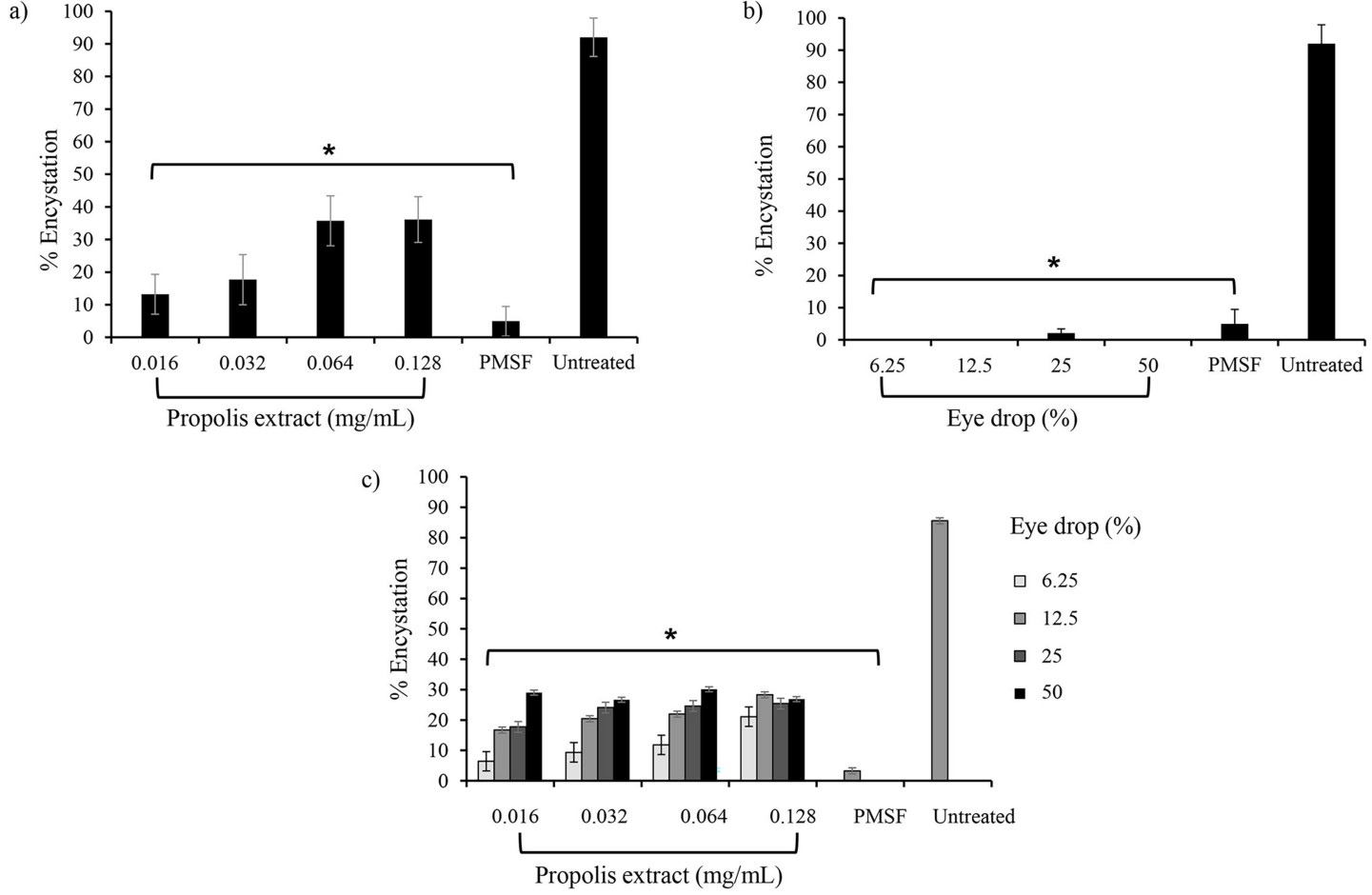

**Figure 1 Effects of propolis extract and eye drops on the encystation of *A. triangularis* trophozoites.** (A) Trophozoites were incubated with various concentrations of propolis extract and (B) eye drops. (C) Effects of combination between propolis extract and eye drops on *A. triangularis* trophozoite encystation. The data represent the mean cell number at 72 h after incubation. Error bars represent the mean ± SD.

### Effect of combination on inhibition of encystation

The percentage of *A. triangularis* encystation was challenged in EDS-3 combination with propolis extract No. 10 at various concentrations. After 72 h, 85.9 ± 4.6% of trophozoites switched to cyst form in the untreated control. The lowest percentage of encystation was seen at 3.6 ± 1.6% and 6.5 ± 2.2% when exposed to PMSF and in EDS-3 (6.25%) combination with propolis extract No. 10 (0.016 mg/mL), respectively compared to the untreated control (Fig. 1C).

### Inhibition of excystation in *A. triangularis*

The effect of propolis extract and EDS treatment on excystation was assessed in PYG medium. Most trophozoites (60.1 ± 2.0%) emerged from cysts when incubated with propolis extract No. 10 at 0.064 mg/mL (1/16 MIC) and the excystation rate decreased (12.2 ± 2.2%) after exposure to high concentrations of propolis extract at 0.512 mg/mL (1/2 MIC) (Fig. 2A). For EDS, the lowest level of excystation (16.6 ± 2.0%) was observed when

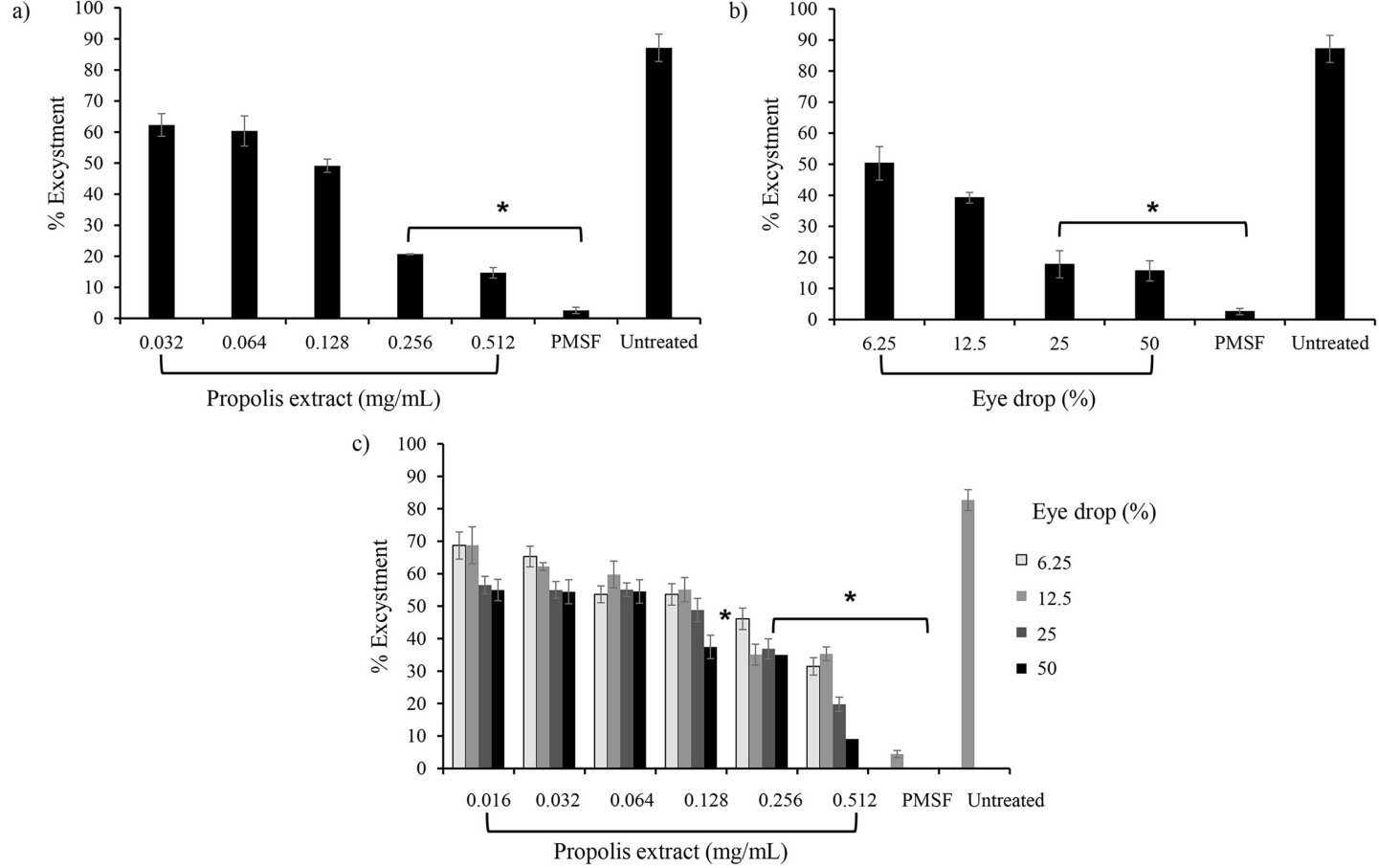

**Figure 2 Effects of propolis extract and eye drops on the excystation of *A. triangularis* trophozoites.** (A) Trophozoites were incubated with various concentrations of propolis extract and (B) eye drops. (C) Effects of combination between propolis extract and eye drops on *A. triangularis* trophozoite excystation. The data represent the mean cell number at 72 h after incubation. Error bars represent the mean ± SD.

treated with 50% (1/2 MIC) of EDS-3 (Fig. 2B). Mature cysts with both layers of the cyst wall were seen as a major cell type in PMSF, propolis extract, and EDS treatment, while trophozoites were observed in the untreated control (Fig. S1).

### Effect of combination on anti-excystation

The combination sets of propolis extract No. 10 and EDS-3 decreased the excystation of *A. triangularis*. The lowest percentage (9.4 ± 4.2%) of excystation was observed after incubation with 0.512 mg/mL of propolis extract No. 10 and 50% of EDS-3 compared to the untreated control (Fig. 2C).

### Effect of combination on anti-amoebic activity

The mean of the percentage of viable trophozoites and cysts is presented in Table 1. There were fewer amoebae in the culture that received propolis extract No. 10, EDS-3 and chlorhexidine than in those treated with chlorhexidine alone under all conditions.

**Table 1 Revised table file.**

| Amoeba stage | Concentration of chlorhexidine (mg/mL) | | | |
|---|---|---|---|---|
| **Trophozoites** | **0.001** | **0.002** | **0.004** | **0.008** |
| Chlorhexidine alone | 69.2 ± 6.9 | 56.4 ± 10.5 | 20.5 ± 7.9 | 5.1 ± 4.0 |
| [1]Combination set of encystation | 33.3 ± 4.0 | 23.1 ± 9.1 | 7.7 ± 0 | 5.1 ± 4.0 |
| [2]Combination set of excystation | 30.8 ± 6.9 | 15.4 ± 0 | 5.1 ± 4.0 | 2.6 ± 4.0 |
| **Cyst** | **0.004** | **0.008** | **0.016** | **0.032** |
| Chlorhexidine alone | 95.0 ± 4.0 | 67.5 ± 0 | 35.0 ± 3.9 | 7.5 ± 0 |
| [1]Combination set of encystation | 40.0 ± 4.0 | 27.5 ± 7.7 | 7.5 ± 0 | 2.5 ± 3.9 |
| [2]Combination set of excystation | 57.4 ± 14.0 | 45.0 ± 6.7 | 5.0 ± 3.9 | 5.0 ± 3.9 |

Notes:
[1] Combination set of encystation includes propolis extract No. 10 (0.016 mg/mL) and EDS-3 (6.25%).
[2] Combination set of excystation includes propolis extract No. 10 (0.512 mg/mL) and EDS-3 (50%).

## Calcofluor white staining

Calcofluor white staining was performed to determine the effects of propolis extract No. 10, and EDS-3 on cellulose synthase of *A. triangularis*. As shown in Fig. 3, calcofluor staining was not observed in trophozoites. Interestingly, 72 h post-encystation or excystation, positive calcofluor (blue color) was observed on cyst surfaces.

## SEM and TEM study

Scanning electron micrographs of the parasite at various stages of encystation after exposure to the combination set of propolis extract No. 10 and EDS-3 are shown in Fig. 4. Trophozoites showed abundant acanthopodia on the surface (Fig. 4A). In the pre-encystation stage, trophozoites were round, which retained the short acanthopodia (Fig. 4B). Mature cysts exhibited an ostiole on the surface with a wrinkled cell wall (Fig. 4C) but some cysts were destroyed by propolis extract and EDS. Cysts exposed to combination set showed the morphological changes (Fig. 4D). At 72 h, loss of extracellular material was observed, which progressed to loss of cellular membrane integrity. The cellular membrane and cytoplasmic damage were evident. Ultrastructural changes within the *Acanthamoeba* cysts were examined using TEM and also confirmed the progressive destruction of *A. triangularis* cysts treated with propolis extract and EDS. Mature cyst of *A. triangularis* were presented in Neff's medium (Fig. 5A), and some treated cysts had smaller size than in the control. Most had rough and thick cyst walls. Inside the cyst wall, there were shrinkage of the cytoplasm with lipid droplets lining the border of the cell. The space between the inner endocyst wall and the cytoplasm was filled (Fig. 5B). Some cells showed significant impairment of morphology. Most cells lost their compactness (Figs. 5C and 5D), ectocyst was no wrinkles (Fig. 5D), and cell was shrunken (Fig. 5E). *In vitro* excystation of *A. triangularis* was induced with propolis extract and EDS. The different stages of excystation were observed. A mature cyst showed a typical morphology which included an ostiole on the surface and a wrinkled cell wall (Fig. 6A). At the pre-emergence stage, the surface topology was indistinguishable from the mature cyst, and its ostioles were clearly observed (Fig. 6B). The remarkable feature of the post-emergence stage is the presence of empty cysts wall and trophozoites in the culture

Encystation | Excystation

Untreated · Combination | Untreated · Combination

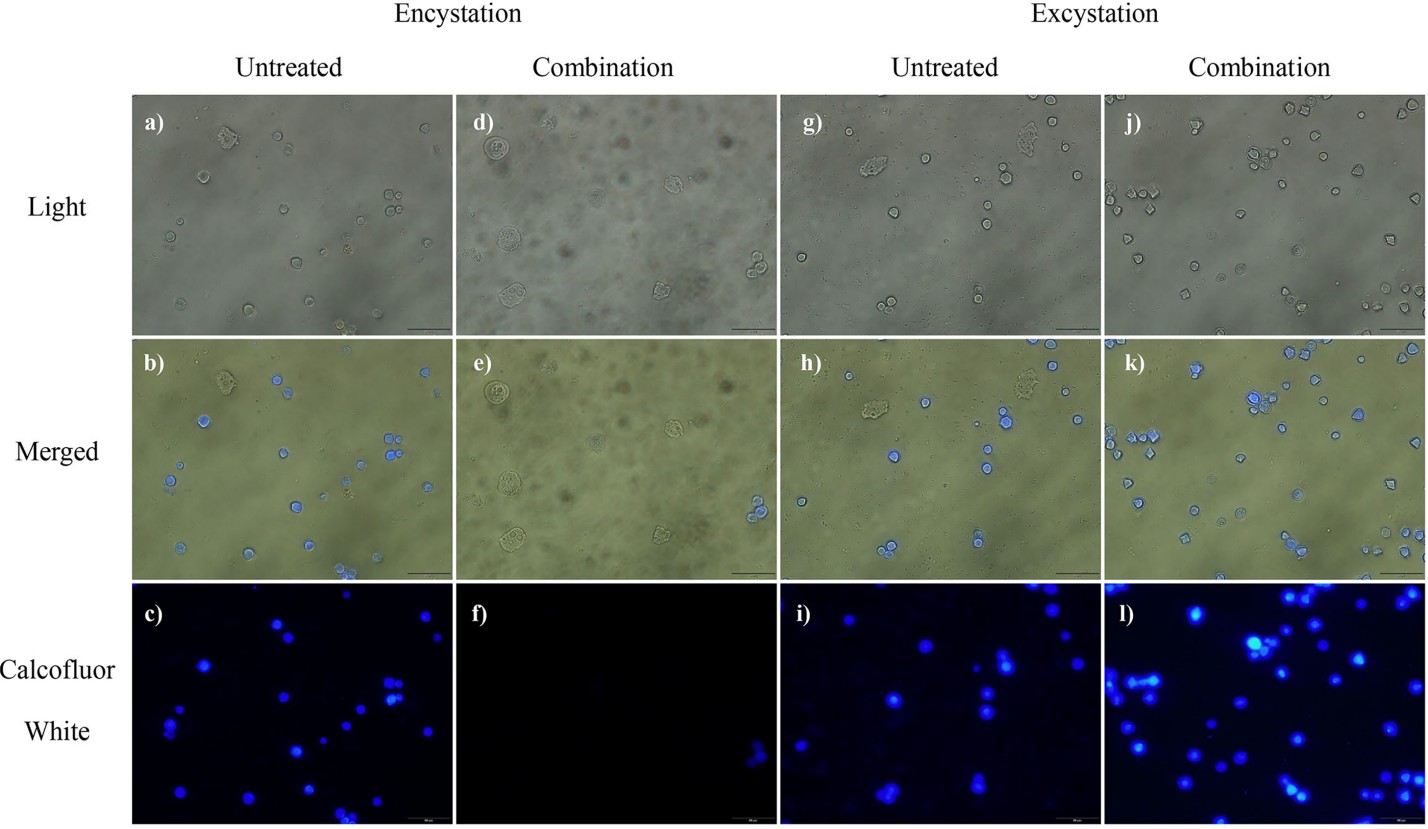

Light
Merged
Calcofluor
White

**Figure 3  Cyst wall formation as detected by calcofluor white staining.** (A–F) At 72 h after the induction of encystment, most of mature cysts were observed in untreated control. In combination, cellulose was not present in trophozoites by calcofluor white staining. (G–L) At 72 h after the induction of excystment, mature cysts remained in both untreated control and in combination. Magnification = 20×.

(Figs. 6C and 6D). Ultrastructural changes within the excysting cysts were examined using TEM. Mature cysts of *A. triangularis* exhibited the typical shape. The ectocyst was distinctly separated from the endocyst by an inter-cystic space, and the endocyst was noticeably separated from the cytoplasmic membrane. Ostioles and operculum were also clearly observed (Fig. 7A). Early stage of excystation, small dense granules were in the region close to the ostiole and associated with the cytoplasmic side of the plasma membrane (Fig. 7B), a less compact operculum (Fig. 7C), and the empty cyst (Fig. 7D) presented in the late phase of excystation. In addition, the irregular features were observed in treated cysts with propolis extract and EDS. The plasma membrane of the excysting cysts have been severely damaged, no defined the nuclear structures, plasm membrane shrunk significantly away from the walls of the endocyst. And there are micellar aggregations inside the cyst suggesting complete plasma membrane destruction (Figs. 7E and 7F).

## Toxicity

At low concentrations, ranging from 0.008–0.064 mg/mL for propolis extract No. 10, the number of live cells was constant over 24 h. However, it had cytotoxic concentration values

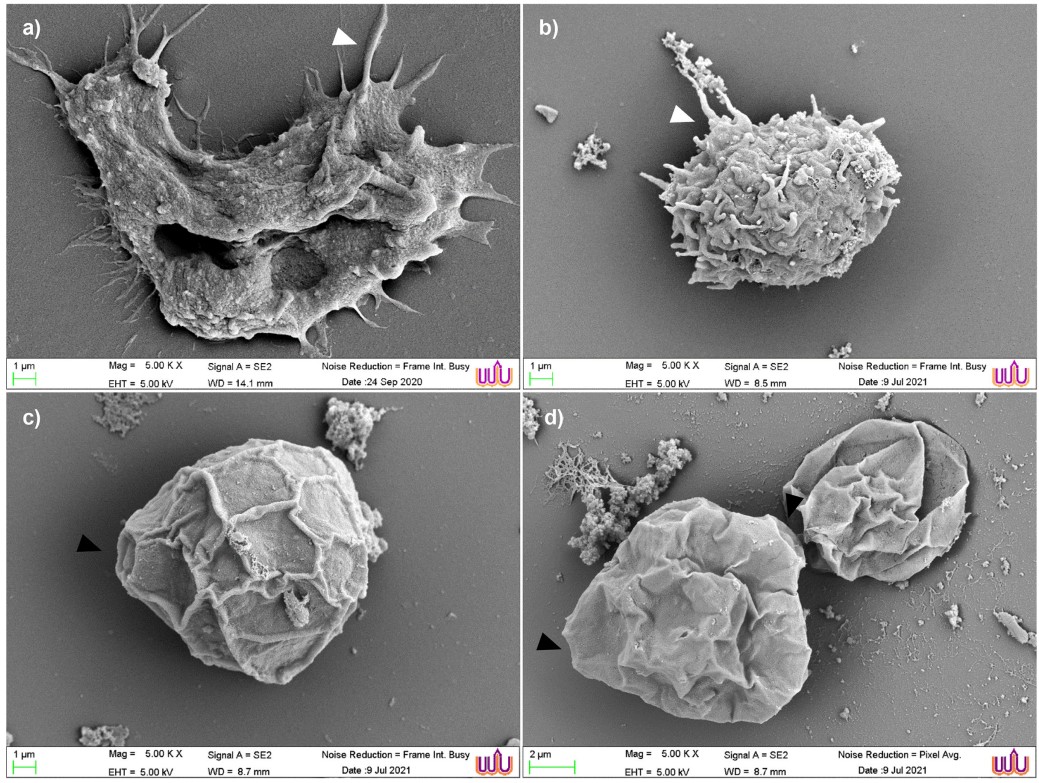

**Figure 4 Scanning electron micrographs of parasite at various stages of encystation after exposure to the combined set of propolis extract and eye drops.** (A) Trophozoite showed abundant acanthopodia on its surface. (B) Pre-encystation stage, trophozoites were round, which remained short acanthopodia. (C) Some mature cyst exhibited an ostiole on the surface which wrinkled cell wall. (D) Destruction cysts by propolis extract and EDS-3.

at 0.128–1 mg/mL that survival of Vero cells lower than 80%. To reduce the toxicity, the combined sets of propolis and EDS-3 were further challenged to determine the survival of Vero cells. A total of 80% of surviving Vero cells were observed at combined concentrations after 24 h of treatment (Fig. S2).

## DISCUSSION

The pharmaceutical activities of natural products have been widely screened because there is a source for finding effective new substances that are cheaper and without side effects. In recent decades, propolis attracted scientific interest due to its biological activities such as anti-inflammatory, anti-proliferative, antioxidant, antiviral, antibacterial, antifungal, and anti-parasitic (*Przybyłek & Karpiński, 2019*). Till date, there are *in vitro* studies of propolis on antiparasitic effect have been reported against *Leishmania* spp., *Trypanosoma* spp., *Plasmodium* spp., *Cryptosporidium* spp., *Giardia* spp., *Toxoplasma gondii*, *Trichomonas vaginalis*, and *Blastocystis* spp. (*Asfaram et al., 2021*). In addition, the cysticidal activity of propolis at 15.62 mg/mL against *Acanthamoeba* spp., has also been reported (*Topalkara et al., 2007*). The anti-*Acanthamoeba* activity of plant and propolis extracts from different cities in Iran was screened in the present study. The highest activity was obtained from the propolis extract No. 10 (Kermanshah City), and the MIC against trophozoites and cysts

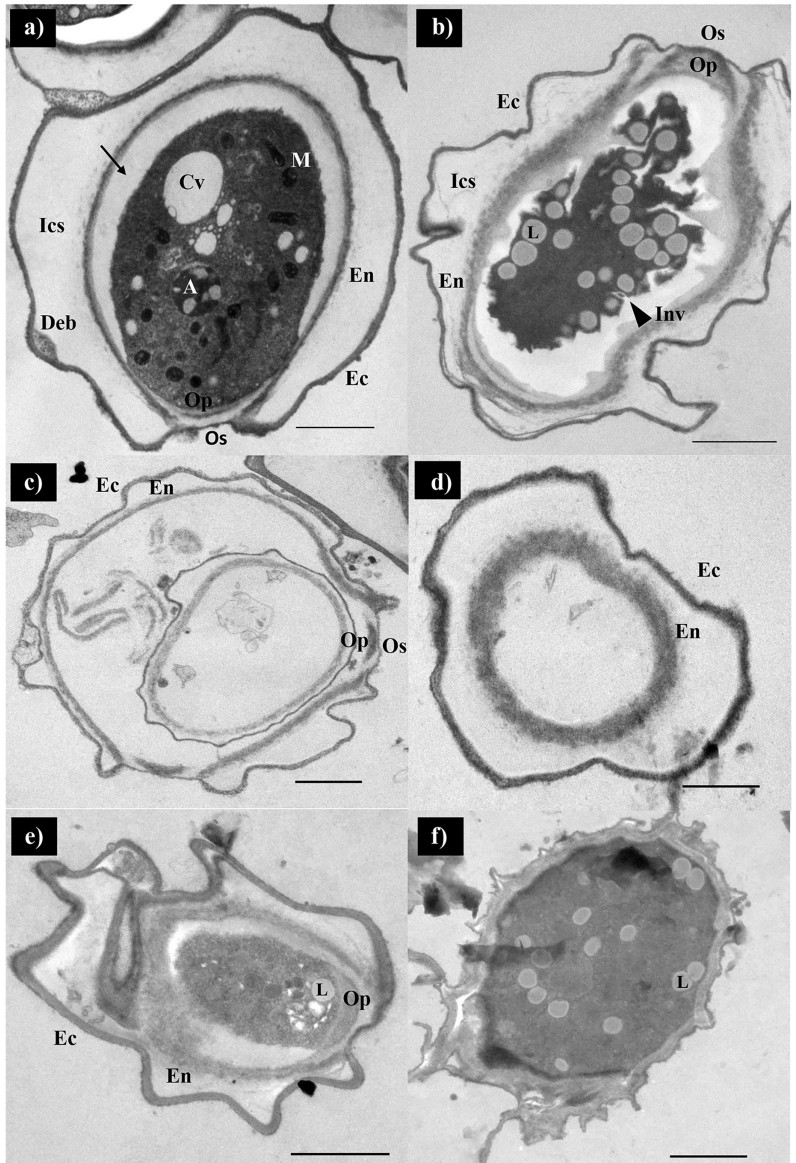

**Figure 5 Transmission electron micrographs of encysting *Acanthamoeba triangularis*.** (A) Ultrathin cross sections of cyst from the mature cyst. (B–F) Incomplete cyst from the treatment with combined set of encystation includes propolis extract (0.016 mg/mL) and EDS-3 (6.25%). Magnifications: A, B, C, E = 3,000×; D, F = 4,000×. Bars: A, B, C, E, F = 2 µM. M, mitochondria; En, endocyst; Ec, ectocyst; Os, ostiole, Op; operculum; A, autolysosome, Cv, contractile vacuole; Ics, inter-cystic space; Deb, deposited cell debris; Inv, invaginations; L, lipid.

was 0.256 and 1 mg/mL, respectively. Based on phytochemical structure as described in our previous study (*Sama-Ae et al., 2022*), flavonoids such as Chrysin, Pinocembrin, and Tectochrysin are the main constituents of propolis extract, and these contain a wide range of biological activities, including anti-parasitic effect. As such, it is not a mere coincidence that similar results were observed in this study. Moreover, the component of propolis can be varied depending on the extraction method, source of plant, and local flora (*Dantas Silva et al., 2017*), which led to the different outputs of propolis samples tested in this study.

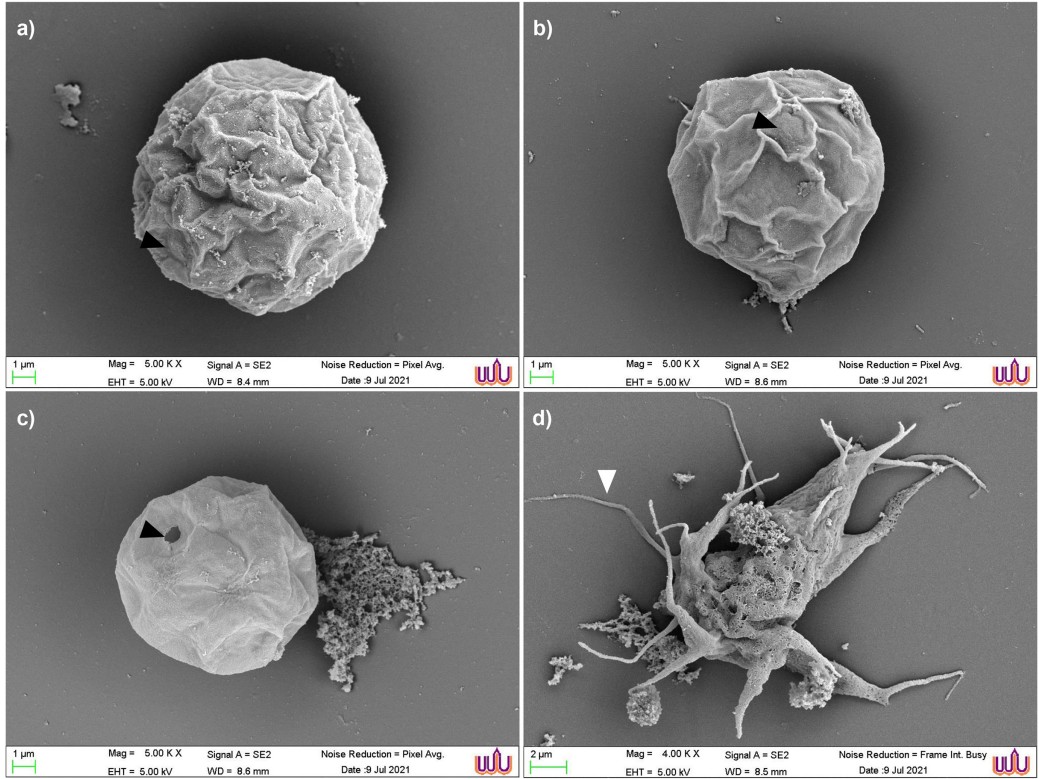

**Figure 6 Parasite at various stages of excystation were exposed after treated with the combination set of propolis extract and eye drops.** (A) A mature cyst showed a typical morphology which included an ostiole on the surface and a wrinkle cell wall. (B) Pre-emergence stage, the surface topology was indistinguishable from the mature cyst, and its ostiole was clearly observed. (C) The remarkable of the post-emergence stage is the presence of empty cyst walls and (D) trophozoites in the culture.

Several studies have applied propolis extracts in cytotoxic investigations. For corneal epithelial cells, propolis extract at higher than 7.81 mg/mL caused cell damage (*Vural et al., 2007*). The safe concentration of propolis extract at 1.4 mg/kg per day was also recommended (*Burdock, 1998*). In this study, different levels of cytotoxicity were found after exposure of Vero cells against propolis extract and combination solutions. However, the cytotoxic effect of the propolis extract at a concentration of at least 0.128 mg/mL was demonstrated against Vero cells. But combination of propolis extract and EDS for inhibition of encystment and excystment were non-toxic to cells. This result indicated that propolis extract was safe when use combination with EDS.

Essentially, the four EDS were included in this study, and EDS-1, -2 and -3 were active against trophozoite, whereas EDS-3 also demonstrated the inhibitory activity against cyst forms. At the same time, EDS-3 was further characterized by its component and benzalkonium chloride of 0.15 mg/mL (0.15%), which was a unique component among the 3 EDS. *Acanthamoeba* trophozoites and cysts were susceptible to benzalkonium chloride, a preservative agent, of 0.05% within 1 and 24 h exposure (*Sunada et al., 2014*). EDS-1 contained 5 mg/mL chloramphenicol which had no cysticidal activity (*Gee Hoon et al., 2011*). While, EDS-2 also contained antibiotics, *i.e.*, neomycin sulfate, gramicidin and

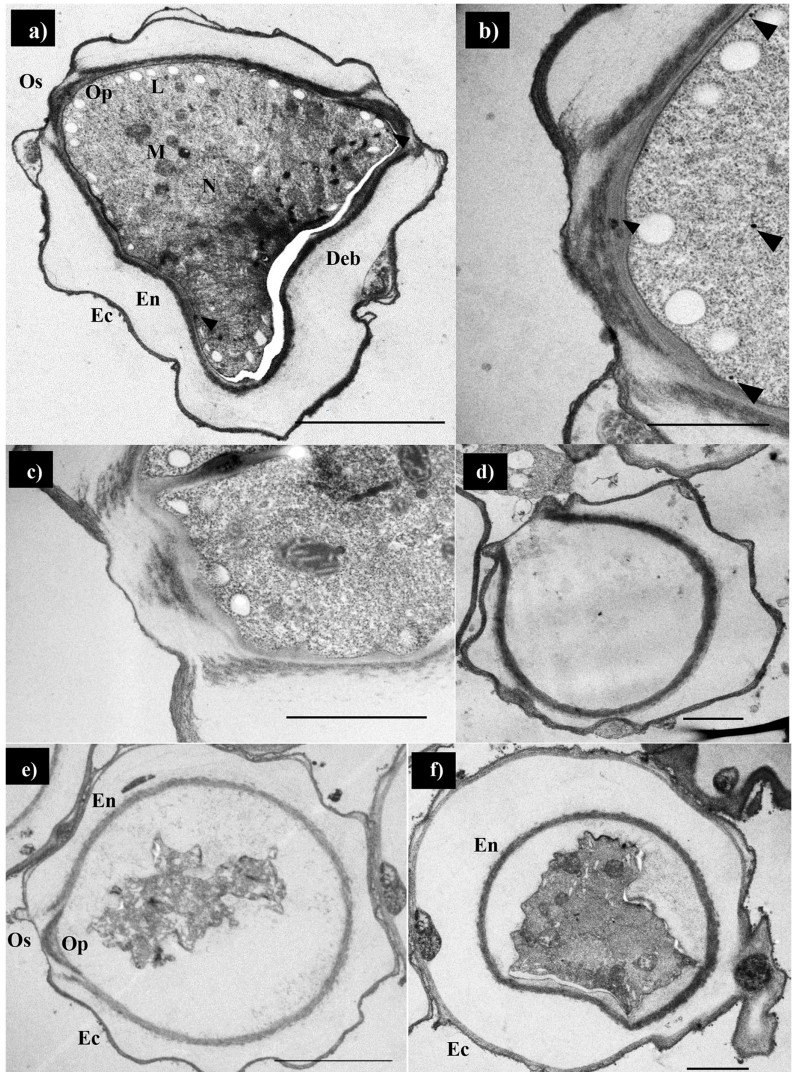

**Figure 7 Transmission electron micrographs of excysting *Acanthamoeba triangularis*.** (A–C) Mature cysts in PYG medium. (A) Small granules are randomly located in the cytoplasm (arrowheads). (B) Some granules appear near the plasma membrane in the ostiole region. (C) The amoeba may form channels by which fibrous components of the cyst wall are ingested. The operculum slowly decayed, and the amoeba forms a pseudopod through the ostiole. (D) The empty cysts are remarkable of post-emergence stage in the culture. (E and F) Alterations in cysts after 72 h treatment with combination set of excystation includes propolis extract (0.512 mg/mL) and eye drops (50%). Magnifications: A = 2,000×; B = 8,000×; C = 10,000×; D = 3,000×; E, F = 2,500×. Bars: D = 2 μM; A, E, F = 5 μM; B, C = 1 μM. M, mitochondria; En, endocyst; Ec, ectocyst; Os, ostiole, Op; operculum; Cv, contractile vacuole; Deb, deposited cell debris.

polymycin, and tested for anti-*Acanthamoeba* activity. However, the cysticidal activity of EDS-2 was also not observed (*Siddiqui, Aqeel & Khan, 2016*). EDS-4 contained 5 mg/mL of levofloxacin which had less effect on *Acanthamoeba* cysts (*Thongseesuksai et al., 2020*). Altogether, this experiment exhibited the anti-amoebic activity of EDS-3, which was shown against both parasite forms. The combination of certain antibiotics with propolis extract indicates another strategy for effective anti-*Acanthamoeba* therapy.

The encystation of *Acanthamoeba* is an essential process for their survival under harsh conditions (*Blanton & Villemez, 1978*). Under these conditions, trophozoites transform into a double-walled cysts form. The outer wall (ectocyst) and inner wall (endocyst) mainly consist of cellulose, D-glucopyranose polymer (*Neff & Neff, 1969*) and an acid-insoluble protein (*Siddiqui & Khan, 2012*). The two cyst walls are normally separated by a space, except the area in which both walls meet and create pores; that is, ostioles occur that are covered by plugs, so-called opercula (*Joo et al., 2020*). However, this double-walled becomes a barrier for drug treatment and leads to clinical drug resistance (*Huang et al., 2017*). Moreover, as often occurs in viable cysts in the corneal stroma after initial successful therapy, they can eventually excyst and lead to re-infection (*Moore et al., 1987*). Thus, the inhibition of encystation process in *Acanthamoeba* enhances the potential of AK treatment. Serine proteinase is a crucial enzyme that produces during the encystation (*Dudley Alsam & Khan, 2008*). The process basically comprises three phases, that is the induction, wall synthesis, and dormancy (*Weisman, 1976*). In this study, *Acanthamoeba* trophozoites were induced to cysts stage by culturing in Neff's medium. PMSF was included as a positive control to inhibit serine proteinase, providing a significant anti-encystation. The data went along with *Leitsch et al. (2010)*, study in which the PMSF inhibited the proteolytic activity at the early stage of encystation. In addition, one of the following: PMSF, propolis extract No. 10, or EDS-3 at low concentration was able to inhibit the encystation of *A. triangularis*. However, the precise underlying mechanism of the anti-encystation by propolis is largely unknown. Other research studies have previously mentioned phenolic compounds such as resveratrol and demethoxycurcumin, which are strong antioxidants with *Acanthamoeba* growth inhibitory effects *in vitro* (*Aqeel et al., 2012*). It raises the possibility that antioxidant activity may be required to inhibit *Acanthamoeba* encystation. Furthermore, *Mahboob et al. (2016)*, reported that other phenolic compounds *i.e.*, ester of caffeic acid and quinic acid, demonstrated the inhibitory effect on encystation by scavenging reactive oxygen species within *Acanthamoeba* cytoplasm. Similarly, the number of cysts formation declined when treated with plant-based secondary metabolites such as betulinic acid, botulin, vanillic acid, and oleanolic acid (*Siddiqui et al., 2022*). Propolis extract No. 10 had flavonoids such as Chrysin, Pinocembrin, and Tectochrysin as major compounds, and our previous study showed the potential ability of these compounds to form hydrogen bonds with the AcSir2 protein that expression is essential for growth and encystation in *A. castellanii* (*Sama-Ae et al., 2022*).

*Acanthamoeba* cysts are living under antibiotic pressure, and this contributes to drug resistance and subsequently results in the recurrence of infection (*Lakhundi, Khan & Siddiqui, 2014*). The excystation occurs when mature cysts are transferred from salt to the growth media (*Chambers & Thompson, 1974*). Like in eye infection, fluids and some microorganisms may provide an appropriate condition to induce excystation of *Acanthamoeba* (*Chávez-Munguía et al., 2005*). In this study, we observed that the high concentration of propolis extract No. 10 or EDS-3 significantly decreased the excystation rate of the *A. triangularis* from cysts to trophozoites. However, it is still not clear of mechanisms involved on how propolis and eye drops inhibited *Acanthamoeba* excystation.

Maslinic acid, a natural triterpene found in olives and propolis, has been shown to inhibit parasitic proteases enzymes (*Zulhendri et al., 2021*). These proteases enzymes are normally secreted within the first 24 h, which may indicate an important role of the enzyme in excystation (*Neal, 1960*). Thereafter, cellulase and chitinase are secreted when excystation is completed, as the two enzymes affect the degradation of empty cyst walls (*Kaushal & Shukla, 1978*).

The current course of AK treatment, chlorhexidine-PHMB combination is recommended for an effective cysticidal activity; however, a long-term treatment of more than 6-months period is required to ensure that the amoeba is completely eradicated. However, prolonged treatment could damage the eyes. Therefore, our prime interest was in the drug combination strategy in order to inhibit or kill the amoeba cysts. Propolis extract-eye drops combination reduced encystation and inhibited excystation. The early symptoms of AK can mimic other types of keratitis that may lead to the misdiagnosis and prolonged therapy which cause *Acanthamoeba* encystation and recurrent infection. Therefore, if an eye drop can eliminate bacteria, fungi and parasites, this may also shorten treatment duration, reduce drug resistance, or encystment in the course of AK. In addition, other drug combinations weakened the cysts, resulting in the killing by chlorhexidine at a lower dose than MIC. Based on the results obtained from this study, these findings are expected to reduce treatment duration in *Acanthamoeba* keratitis. In addition, these findings suggest that combination holds promise in the improved treatment and management of *Acanthamoeba* as to reduce the rate of drug use and drug resistance of *Acanthamoeba*. In the future investigations the antiparasitic activity of the fractions from propolis extract will be evaluated, to provide more insights on their effectiveness for the treatment of AK.

## CONCLUSIONS

A promising natural product with *Acanthamoeba* growth-inhibitory effect and low toxicity would be an ideal property that can be developed for AK treatment. Regarding the pharmacological activities of propolis extract No. 10 against both trophozoites and cysts, thus, its therapeutic potential may be considered. However, our study was conducted with a crude extract of propolis No. 10 that warrants further comprehensive investigations on the anti-*Acanthamoeba* activity of active compounds within the propolis extract No. 10 and the mechanism of encystation or excystation in response to this potential extract. Cytotoxicity is a limiting factor for the use of propolis extract No. 10 so that propolis extract at high concentrations (0.128–1 mg/mL) had anti-*Acanthamoeba* activity and damaged cysts. The study of the important mechanisms of the pure compounds in propolis extract No. 10 or combination the extract with drugs such as chlorhexidine is another way to reduce the toxic of the extract or chlorhexidine.

## ACKNOWLEDGEMENTS

We highly appreciate for the support of the Research Institute of Health Science (RIHS) at Walailak University for the laboratory facilities and Assoc. Prof. Dr. Chuchard Punsawad, School of Medicine, Walailak University for providing the human Vero cell line.

### Funding

This research was financially supported by the Royal Patronage of Her Royal Highness Princess Maha Chakri Sirindhorn—Botanical Garden of Walailak University, Nakhon Si Thammarat (Grant No. RSPG-WU-26/2566 and 14/2567) and the School of Allied Health Science, Walailak University, Thailand (Grant No. WU-SAH 0005/2023). The funders had no role in study design, data collection and analysis, decision to publish, or preparation of the manuscript.

### Grant Disclosures

The following grant information was disclosed by the authors:
Royal Patronage of Her Royal Highness Princess Maha Chakri Sirindhorn—Botanical Garden of Walailak University, Nakhon Si Thammarat: RSPG-WU-26/2566
and 14/2567.
School of Allied Health Science, Walailak University, Thailand: WU-SAH 0005/2023.

### Competing Interests

Sonia M. R. Oliveira is an Academic Editor for PeerJ.

### Author Contributions

- Suthinee Sangkanu performed the experiments, analyzed the data, prepared figures and/or tables, and approved the final draft.
- Abolghasem Siyadatpanah performed the experiments, analyzed the data, authored or reviewed drafts of the article, and approved the final draft.
- Roghayeh Norouzi performed the experiments, authored or reviewed drafts of the article, and approved the final draft.
- Julalak Chuprom performed the experiments, prepared figures and/or tables, and approved the final draft.
- Watcharapong Mitsuwan performed the experiments, analyzed the data, prepared figures and/or tables, and approved the final draft.
- Sirirat Surinkaew performed the experiments, analyzed the data, authored or reviewed drafts of the article, and approved the final draft.
- Rachasak Boonhok performed the experiments, prepared figures and/or tables, authored or reviewed drafts of the article, and approved the final draft.
- Alok K. Paul analyzed the data, prepared figures and/or tables, and approved the final draft.
- Tooba Mahboob conceived and designed the experiments, analyzed the data, authored or reviewed drafts of the article, and approved the final draft.
- Imran Sama-ae conceived and designed the experiments, performed the experiments, authored or reviewed drafts of the article, and approved the final draft.
- Sonia M. R. Oliveira conceived and designed the experiments, analyzed the data, authored or reviewed drafts of the article, and approved the final draft.

- Tajudeen O. Jimoh analyzed the data, prepared figures and/or tables, and approved the final draft.
- Maria de Lourdes Pereira conceived and designed the experiments, authored or reviewed drafts of the article, proof read of manuscript, language editing, and approved the final draft.
- Polrat Wilairatana conceived and designed the experiments, authored or reviewed drafts of the article, and approved the final draft.
- Christophe Wiart analyzed the data, authored or reviewed drafts of the article, and approved the final draft.
- Mohammed Rahmatullah conceived and designed the experiments, authored or reviewed drafts of the article, and approved the final draft.
- Monvaris Sakolnapa performed the experiments, prepared figures and/or tables, and approved the final draft.
- Veeranoot Nissapatorn conceived and designed the experiments, analyzed the data, authored or reviewed drafts of the article, and approved the final draft.

## Data Availability

The raw data for Figs. 1 and 2 and Table 1 are available in the Supplemental Files.

## Supplemental Information

Supplemental information for this article can be found online at http://dx.doi.org/10.7717/peerj.16937#supplemental-information.

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
