# Peer review of "Efficacy of propolis extract and eye drop solutions to suppress encystation and excystation of *Acanthamoeba triangularis* WU19001-T4 genotype"

_PeerJ, doi:10.7717/peerj.16937_

## Round 0.1 · original submission · Major Revisions

Dear authors,

Thank you for your patience while your manuscript was peer-reviewed at PeerJ. It has now been evaluated by the editor and by three independent reviewers with relevant expertise.

I invite you to respond to the reviewers' detailed comments and revise your manuscript. All the reviewers' comments need to be addressed before the manuscript can be accepted.

Thank you for submitting your manuscript to PeerJ and I look forward to receiving your revision.

Best,

Xiaotian Tang, PhD
Academic Editor, PeerJ

**Language Note:** The review process has identified that the English language must be improved. PeerJ can provide language editing services - please contact us at [email protected] for pricing (be sure to provide your manuscript number and title). Alternatively, you should make your own arrangements to improve the language quality and provide details in your response letter. – PeerJ Staff

Reviewer 1 ·

Basic reporting

The present study provides valuable information about the potential activity of Propolis extract and eye drop solutions against encystation and excystation of Acanthamoeba triangularis WU19001-T4 genotype.
However, there is a major concerns to be considered for publications.
1- The entire manuscript requires complete rewriting since English language requires an extensive editing.
2- The introduction is empty and required to be extended. For example, the epidemiological profile of the parasite and the potential in in vivo activity of Propolis against various parasites such as Toxoplasma gondii should be discussed in details, among others.
3- Research question and aim of the study is unclear. It should be addressed.

Experimental design

1-Research question is not well defined,
2-please do consider dividing your methodology into several sub sections and Gas chromatography-mass spectrometry (GC-MS) analysis should be moved directly after materials. You should perform this step before nay further biological studies.
3- Why did not you study the possible mechanisms underlying this activity?

Validity of the findings

1-The limitation of your study should be included at the end of discussion section.
2-Conclusions should be rewritten to present your own findings and suggestions of future perspectives.

Additional comments

Extensive English language editing is required.

·

Basic reporting

1. The structure is correct.

2. It is essential to enhance the clarity of your text in the English language to make it easily understandable for an international audience. Certain areas that could be improved include lines 78, 321, 324, 365, and 388, as the current wording poses challenges for comprehension. I recommend having a colleague who is proficient in English and knowledgeable about the subject matter review your manuscript or consider reaching out to a professional editing service. This way, you can ensure that your message is conveyed effectively to readers worldwide.

3.Your introduction lacks sufficient detail. I recommend enhancing the explanation in lines 111-112 to offer stronger justification for your study. Specifically, you should elaborate on the knowledge gap your research addresses, why eye drops are necessary during early treatment, and why their improvement is essential, leading you to propose the use of propolis.

3. In line 116, the authors should present the research group's recent advancements concerning the interaction of Acanthamoeba castellani proteins with propolis extracts, as these findings form the basis for further testing it with A. triangularis.

4. Figures are relevant to the content of the article, of sufficient resolution. Figures description should be improved for better understanding, for example in Figure 1 “(a) Trophozoites were incubated with various concentrations of Propolis extract and (b) eye drops. (c) Effects of combination between Propolis extract and eye drops on A. triangularis trophozoite encystation. The data represent the mean cell number at 72 hours after incubation. Error bars represent the mean ± SD” I would change to: “Percentage of encystation of A. triangularis trophozoites 72h after incubation a) propolis extract, b) eye drops, c) combination of propolis extract and eyedrops. Untreated cells and PMSF treated cells were used as controls. Error bars represent the mean ± SD. * p<0.05”

3. Arrows should be included in the SEM images to emphasize the effects on specific structures, such as the ostiole, achantopodia, etc.

4. The TEM figure captions should clearly indicate which images represent the control group and specify the time points when the photos were taken (a-d).

5. All appropriate raw data have been made available in accordance with Data Sharing policy.

6. 1. Original primary research within Aims and Scope of the journal.
The relevance of the study is not clear. The abstract lacks a clear explanation of the knowledge gap; the introduction suggests the need for a different treatment due to the long-term development of resistance and secondary effects in the eye. Furthermore, in the discussion, it is emphasized that a more cost-effective and efficient treatment is required. Therefore, it is essential to highlight the same relevance in those respective sections.
2. Rigorous investigation performed to a high technical & ethical standard.
It is imperative to present the analyses conducted to determine encystment and ensure that all studies were performed using the protozoan in its mature cyst form rather than precysts or trophozoites. These analyses may include techniques such as immunofluorescence, flow cytometry, selection for water resistance, recovering the trophozoites from the supernatant, and the cysts were scraped from the surface, etc.

3. Methods described with sufficient detail & information to replicate.
Methods should be described with sufficient information to be reproducible by another investigator.
1. Line 134: The method of plant identification is unclear. If it was based on morphological characteristics, please provide the herborization number. If molecular methods were used, include the relevant methodology and results.
2. Table S1: Clarify the collection locations of the plants. Reorganize the table to include "city" and "coordinates" columns adjacent to each plant entry.
3. Line 135: Describe the selection process of plant material, specifying the plant part used, whether plants with lesions or fungi were eliminated. Additionally, include details on how the plant material was ground, whether it was previously dried, and the equipment used for grinding, such as a mortar or blender. Mention whether the powder was sifted.
4. Provide a justification for using these eye drops, as they are not the conventional treatment for this protozoan infection.
5. Justify the use of PMSF as a control in the encystment and de-encystment tests instead of chlorhexidine, which is the treatment of choice.
6. Line 251: Specify the concentration of the propolis stock used or the final concentration in the experiment. Also, mention the solvent in which it was dissolved.
7. In the cytotoxicity assay, provide a rationale for using Vero cells, as they are not the target cells for the treatment, which would be corneal cells. Additionally, justify the use of cells from a different species (Vero cells) than the target (human).
8. Include the viability results of cells treated only with the eye drops.
The draft contains experiments that are unrelated to the stated objective in the introduction. These irrelevant experiments involve the use of plant extracts, and it is unclear why they were conducted or how they relate to testing their activity against Acanthamoeba, as no follow-up is provided. The stated objective in the introduction is to evaluate amoebicidal activity and anti-Acanthamoeba encystation or excystation using Propolis extract and eye drops as an alternative treatment strategy for Acanthamoeba infection. Therefore, I suggest removing the section on plant extracts and their anti-acanthamoeba activity, as well as adjusting the article title; an alternative could be conducting MIC, encystation and excystation, calcofluor white staining, SEM, and TEM experiments with the plant extracts.

Experimental design

1. The relevance of the study is not clear. The abstract lacks a clear explanation of the knowledge gap; the introduction suggests the need for a different treatment due to the long-term development of resistance and secondary effects in the eye. Furthermore, in the discussion, it is emphasized that a more cost-effective and efficient treatment is required. Therefore, it is essential to highlight the same relevance in those respective sections.

2. It is imperative to present the analyses conducted to determine encystment and ensure that all studies were performed using the protozoan in its mature cyst form rather than precysts or trophozoites. These analyses may include techniques such as immunofluorescence, flow cytometry, selection for water resistance, recovering the trophozoites from the supernatant, and the cysts were scraped from the surface, etc, whatever you used to determined this.

3. Line 134: The method of plant identification is unclear. If it was based on morphological characteristics, please provide the herborization number. If molecular methods were used, include the relevant methodology and results.

4. Table S1: Clarify the collection locations of the plants. Reorganize the table to include "city" and "coordinates" columns adjacent to each plant entry.

5. Line 135: Describe the selection process of plant material, specifying the plant part used, whether plants with lesions or fungi were eliminated. Additionally, include details on how the plant material was ground, whether it was previously dried, and the equipment used for grinding, such as a mortar or blender. Mention whether the powder was sifted.

6. Provide a justification for using these eye drops, as they are not the conventional treatment for this protozoan infection.

7. Justify the use of PMSF as a control in the encystment and de-encystment tests instead of chlorhexidine, which is the treatment of choice.

8. Line 251: Specify the concentration of the propolis stock used or the final concentration in the experiment. Also, mention the solvent in which it was dissolved.

9. In the cytotoxicity assay, provide a rationale for using Vero cells, as they are not the target cells for the treatment, which would be corneal cells. Additionally, justify the use of cells from a different species (Vero cells) than the target (human).

10. Include the viability results of cells treated only with the eye drops.

Validity of the findings

The conclusion requires rewriting to establish a clear link with the initial research question, drawing upon the obtained results.

Additional comments

The abstract requires improvement.
1. It remains to be argued why propolis and the eye drops used in the study should be used as an alternative to treat AK.
2. Methodology is incomplete. Please mention the encystation, excystation, calcofluor white stain, TEM and SEM methods.
3. In Results you mention “In combined sets of Propolis extract and eye drops, they showed slightly increasing inhibition of encystation and excystation”, you should specify the concentration were you observed this phenomenom.

4. You should explain more the TEM and SEM results.

The discussion section requires improvement in the following areas:

1. In Lines 421-425, although they mention the cytotoxic effect of propolis on Vero cells, they do not provide an in-depth analysis of the underlying reasons or the potential implications of this finding.

2. Regarding Line 463, the compounds mentioned to have an effect against Acanthamoeba differ from those identified in this study. A more valuable discussion would center on the specific effect of the propolis compounds identified in this research.

3. There is a notable absence of discussion about the potential benefits and outcomes of combining propolis with eye drops.

4. It is of utmost importance to thoroughly discuss the results obtained from calcofluorine white staining, as well as the observations made in TEM and SEM, as these insights are instrumental in understanding the mechanism of action of propolis against Acanthamoeba triangularis.

·

Basic reporting

Clear and unambiguous, professional English used throughout.

It is an interesting manuscript that studies the effect of Propolis extract-eye drops combination in the encystation reduction and excystation inhibition of Acanthamoeba triangularis.
The manuscript is clear and well-written.

Literature references, sufficient field background/context provided.

The manuscript was written with appropriate references with some exceptions in the methodology, results, and discussion sections.
Regarding the methodology of calcofluor white staining, Moon et al. (2014) described in detail this methodology. Authors may consider adding this reference to the list.
Line 431, the authors must review the concentration of Benzalkonium chloride as well as the exposure time of Benzalkonium chloride reported by Sunada et al. (2014) with amoebicidal and cysticidal activities.
Line 479, Kaushal and Shukla described the participation of cellulase and chitinase during the excystment of Hartmannella culbertsoni in 1978, not in 1977.
The result of the gas chromatography-mass spectrometry analysis of Propolis- Kermanshah extract was included as part of a previous publication (Sama-Ae et al., 2022). Authors may consider adding this reference to the list.
The authors should include in the discussion section information about their previous in silico studies about chrysin, tectochrysin, and pinocembrin as possibly responsible for encystment and excystment inhibition of Acanthamoeba (Sama-Ae et al., 2022).

Professional article structure, figures, tables. Raw data shared.

The manuscript shows a professional article structure, the results of the antiparasitic and cytotoxic activity of the Propolis extract-EDS combination are presented in tables and graphics.
Figures are selected appropriately and show the ultrastructural analysis of the effect of Propolis extract- eye drop solution combination on the encystment and excystment of A. triangularis. However, some observations related to the figures are mentioned below:
The effect of Propolis extract and eye drop solution (EDS) on the encystation and excystation of A. triangularis is described in Figures 1 and 2 respectively, however, the authors should mention which is the propolis extract-city evaluated as well as the number of EDS.
In Figure 2 (excystment process) concentrations used of Propolis extract: 0.032 mg/mL (1/32 MIC) (Fig2 a and c) and 0.016 mg/mL (1/64 MIC) (Fig 2c) are described; but this information should also be given in the methodology section.
Concentrations of Propolis extract and EDS combination employed should be described both in Figure 3 and in the methodology section (cyst wall formation).
Scanning and transmission electron micrographs of encysting A. triangularis show sufficient resolution (Figure 4 and 5, respectively), however, authors need to point out in the figures the observed effect (using arrows, arrowheads, asterisks). Besides, the authors should include the concentrations used in the Figure 4 caption.
In Figure 5a, is the mature cyst of untreated control parasites described?
Capital letters (D-F) in the Figure 5 caption are not correlated with labels on the micrographs. Moreover, the unit of scale bar must be indicated in micrometers.
Scanning and transmission electron micrographs of excysting A. triangularis show sufficient resolution (Figure 6 and 7, respectively), however, authors should indicate using different labels the effect described. Besides, the authors should include the figure caption description using the same marks, as well as the concentrations used in Figure 6.
Figure 7, the correct abbreviation of micrometers is required in the figure caption. Additionally, the abbreviation for contractile vacuoles (Cv) is not labeled in the micrograph.
Figure S1 mentions Propolis19, what does the number 19 refer to?
On the other hand, raw data of results are included; however, some inconsistencies were found.
The percentages of A. triangularis viability reported in Table 1 show inconsistencies with raw data (excel) including the standard deviation values.
Raw data of the effect of Propolis extract at 0.032 mg/mL and in EDS combination at 0.016 mg/mL on the parasite excystation are not included in the Excel file.

Self-contained with relevant results to hypotheses.

Amoebicidal and cysticidal properties of the Propolis extract-EDS3 combination were demonstrated and this combined therapeutic could offer an alternative for AK treatment

Experimental design

Original primary research within Aims and Scope of the journal.

The research of this manuscript is considered within the scope of the journal in Biological Sciences, the methodology employed is not a novelty, however, the proposal of the use of a naturally occurring substance in combination with eye drop solutions to suppress encystation and excystation of A. triangularis is interesting. Even so, these results hold the promise of finding a new therapeutic alternative for the treatment of Acanthamoeba keratitis.

Research question well defined, relevant & meaningful. It is stated how research fills an identified knowledge gap

The research question of the present study is well-defined. To evaluate the antiparasitic activity of natural extracts in combination with commercial eye drop solution, as well as cytotoxic studies in mammalian cells line, authors employed in vitro cell culture and electron microscopy techniques. Further investigations to evaluate the antiparasitic activity of the individual fractions of the Propolis extract will provide more insights into their potential use for the treatment of AK. It is also of interest to search for its mechanism of action.

Rigorous investigation performed to a high technical & ethical standard.

In the present study, different biological assays were performed to characterize the antiparasitic effect of Propolis extract in combination with eye drop solution-3. The experiments were performed in triplicate and data were expressed as mean ± SD.
There are some technical observations mainly related to the Methodology and Results sections
The authors should include the percentages of the viability of the parasites treated with eye drop solutions (EDS 1-4) in Table S3.
Table S4 should only indicate the eye drop solutions' MIC values, as their composition is already mentioned in Table S2. Additionally, the standard deviation in the MIC values should be added.
In line 238 the correct abbreviation of nanometers is required.
Line 293, how did the authors calculate the viability decrease in cyst and trophozoite treated with ethanolic extract?
Figure S2. The reduction of viability (%) of Vero cells treated with 0.512 mg/mL Propolis extract is significant, however in combination with EDS-3 there is no cytotoxic at the same concentration, could authors explain this effect?
Line 458, the genus and species names should be in Italics

Methods described with sufficient detail & information to replicate

Antiparasitic in vitro evaluation and ultrastructural analysis by scanning and transmission electron microscopy were used to demonstrate the effect of Propolis extract-EDS3 combination on the encystation reduction and excystation inhibition of A. triangularis, in addition, cytotoxicity assays in Vero cells and the chemical composition of Propolis-Kermanshah extract was analyzed.
Some other observations related to the Methodology:
Line 172, the authors must include how many concentrations were used to calculate the Minimum inhibitory concentration (MIC) and mention that chlorhexidine was used as a positive control. Chlorhexidine was acquired through organic synthesis or commercial presentation?
In the electron microscopy methodology, authors should mention the concentration of the combination of EDS-propolis extract evaluated.
Although human corneal epithelial cells will be optimum for this study, Vero cells line is validated for toxicological evaluation. However, the value of the 50% cytotoxic concentration (CC50) of Vero cells treated with Propolis extract alone or in combination with EDS-3 should be included.

Validity of the findings

Impact and novelty not assessed.

Further investigations to evaluate the anti-Acanthamoeba activity of the individual fractions of the Propolis-Kermanshah could provide more detailed information about their antiparasitic activity and their mechanism of action.
Studies about chrysin, tectochrysin, and pinocembrin as possibly responsible for encystment and excystment inhibition of A. triangularis.
In vivo studies are required to be able to propose a propolis-Kermanshah- EDS-3 combination as an alternative therapy for the treatment of Acanthamoeba keratitis.

Meaningful replication encouraged where rationale & benefit to literature is clearly stated.

A comparison of the results obtained with those reported in the literature was included in the discussion section however the authors can provide more information in this respect:
Line 430, is there any evidence of antiparasitic activity of Antazoline and Tetrahydrozoline?
Previous studies by Siddiqui et al. (2016) demonstrated that a combination of neomycin plus polymyxin B plus bacitracin exhibited amoebistatic and amoebicidal effects as well as cysticidal properties, however in the present study cysticidal activity of EDS-2 was not observed. Could authors discuss this?

All underlying data have been provided; they are robust, statistically sound, & controlled

Biological replicates with standard deviation are presented in table 1 and S3, Figures 1-2 and S2, with the exception of table S4.

Conclusions are well stated, linked to original research question & limited to supporting results

Amoebicidal and cysticidal activities of Propolis extract were demonstrated and its effect in the encystment and excystment of the parasite in combination with EDS-3 was evident however, importantly the authors should mention that the propolis-Kermanshah city was employed in this research

Additional comments

'no comment'

---

## Round 0.2 · Minor Revisions

The manuscript has been significantly improved. However, there are few edits as suggested by Reviewer 3. Please revise them.

·

Basic reporting

The comments were addressed appropriately

Experimental design

The comments were addressed appropriately

Validity of the findings

The comments were addressed appropriately

Additional comments

There are no comments

·

Basic reporting

no comment

Experimental design

no comment

Validity of the findings

no comment

Additional comments

In Table S3 and Table S4, the numbers of propolis extract should be included, for example, No. 1 (Tabriz city), etc.
In different sections of the manuscript, the number of Propolis extracts (No 10) is mentioned, in other cases, Propolis-Kermanshah city or simply Propolis extract, the authors can unify.
The abbreviation for contractile vacuoles (Cv) is not labeled in the micrograph but is included in figure 7 caption
In the SEM figure 4 and 6 captions, the arrowhead label does not describe the meaning. However, is described in the response to the reviewers.
The description of the black arrowhead and black arrow in the Figure 5 caption is required
Is the number 19 labeled in Figure S1 correct?

---

## Round 0.3 · accepted · Accept

Dear Authors,

I am pleased to inform you that your article, "Efficacy of Propolis extract and eye drop solutions to suppress encystation and excystation of Acanthamoeba triangularis WU19001-T4 genotype", has now been accepted for publication in PeerJ. Congratulations!

Thank you for your submission and we hope you will continue to support PeerJ.

Best,

Xiaotian Tang